# Possible Biomarkers for Cancer Immunotherapy

**DOI:** 10.3390/cancers11070935

**Published:** 2019-07-03

**Authors:** Takehiro Otoshi, Tatsuya Nagano, Motoko Tachihara, Yoshihiro Nishimura

**Affiliations:** Division of Respiratory Medicine, Department of Internal Medicine, Kobe University Graduate School of Medicine, Kobe, Hyogo 650-0017, Japan

**Keywords:** cancer immunotherapy, programmed cell death-ligand 1 expression, tumor mutational burden, neoantigens, mismatch repair status, specific gene mutations, gut microbiome

## Abstract

Immune checkpoint inhibitors (ICIs) have drastically changed the clinical care of cancer. Although cancer immunotherapy has shown promise in various types of malignancies, thus far, the proportion of patients who can benefit from ICIs is relatively small. Immune-related adverse events and high cost are unavoidable problems. Therefore, biomarkers defining patients that are most likely to benefit from ICIs are urgently needed. The expression of programmed cell death-ligand 1 (PD-L1) is a logical biomarker for the prediction of response to anti-PD1/PD-L1 immunotherapies. However, its usefulness is currently debatable because of its varied definition, threshold, and spatial/temporal heterogeneity. Recently, it was reported that the tumor mutational burden, expression of neoantigens, mismatch repair status, and specific gene mutations may be markers for the success of treatment with ICIs. Moreover, it was suggested that the fecal microbiota prior to immunotherapy may play an important role in predicting the efficacy of ICIs. In this review, we focused on these potential biomarkers for cancer immunotherapy reported in recent clinical articles. Further studies are warranted to develop a predictive model using these biomarkers, with the aim of practicing precision medicine in cancer immunotherapy.

## 1. Introduction

Immune checkpoint inhibitors (ICIs) prevent inhibitory signals in T cells and restore T-cell immune activity. Anti-cytotoxic T-lymphocyte-associated antigen 4 (CTLA4), anti-programmed cell death 1 (PD1), and anti-PD1-ligand 1 (PD-L1) monoclonal antibodies have drastically changed the clinical care of cancer. In 2011, ipilimumab—an anti-CTLA4 antibody—was approved by the Food and Drug Administration (FDA) for anticancer treatment. It demonstrated a positive impact on the overall survival (OS) of patients with metastatic melanoma after several years of follow up [1,2]. Subsequently, pembrolizumab—an anti-PD1 antibody—provided better OS than ipilimumab in advanced melanoma patients [3]. Furthermore, pembrolizumab showed significantly better progression-free survival (PFS) and OS versus platinum-based chemotherapy in patients with advanced non-small cell lung cancer (NSCLC) that lack targetable mutations [4]. At present, PD-1/PD-L1 inhibitors have been approved by the FDA for numerous types of cancer, including melanoma, NSCLC, head and neck squamous cell cancer, urothelial cancer, liver cancer, Hodgkin’s lymphoma, renal cell cancer (RCC), gastric cancer, and colorectal cancer [5]. However, the proportion of patients who can benefit from ICIs is relatively small. For instance, only 20–25% of NSCLC patients showed a sustainable response to ICIs [6]. Therefore, the discovery of clinical or biological biomarkers is urgently warranted. The aim of this approach is to practice precision medicine in cancer immunotherapy. Moreover, the clinical use of ICIs is also influenced by cost-effectiveness. Therefore, establishing useful biomarkers of response to ICIs is additionally important to avoid financial burden. In this review, we focus on the potentially useful biomarkers in this field.

## 2. The Expression of PD-L1 as a Predictive Biomarker

The expression of PD-L1 on normal tissues is limited. However, numerous tumor cells overexpress PD-L1 as a strategy to evade immune responses [7]. Thus, the expression of PD-L1 on tumor cells may play an important role in suppressing T-cell immune activity. In a previous study involving 42 patients who had received anti-PD-1 antibody (i.e., 18 with melanoma, 10 with NSCLC, 7 with colorectal cancer, 5 with renal cell cancer, and 2 with prostate cancer), PD-L1-positive tumors exhibited significantly better objective response versus PD-L1-negative tumors [8]. A subsequent study also showed that the expression of PD-L1 in ≥50% of tumor cells was associated with better efficacy of pembrolizumab in patients with advanced NSCLC [9]. In other studies that included patients with metastatic RCC or recurrent head and neck cancer treated with nivolumab, it was shown that the level of PD-L1 expression in tumors may predict the efficacy of nivolumab [10,11]. However, nivolumab showed its benefit among patients with advanced RCC, irrespective of the expression of PD-L1 [12]. In addition, in previously treated patients with NSCLC, treatment with atezolizumab resulted in a clinically relevant improved OS versus docetaxel, regardless of the expression of PD-L1 [13]. These results suggest that the treatment effect of anti-PD-1/PD-L1 treatment is independent of the expression of PD-L1 [6]. The result of each study mentioned above is summarized in Table 1.

There are several possible explanations for these contradictory results. Firstly, various studies have used different definitions of PD-L1 expression. Thus, we should pay attention to the cell type on which PD-L1 is expressed. For example, Tang et al. found that PD-L1 in host myeloid cells was essential for the response to immune checkpoint blockade, whereas PD-L1 on tumor cells was mostly dispensable for the response [7]. In another study including patients with various types of cancers (*n* = 175), the association between response to treatment with ICIs and expression of PD-L1 on tumor-infiltrating immune cells reached statistical significance. However, the association between response and expression of PD-L1 on tumor cells did not reach statistical significance [14].

Secondly, different thresholds of PD-L1 expression may lead to variable results. For example, in a study evaluating the efficacy of pembrolizumab plus chemotherapy in patients with metastatic NSCLC, the predictive effect of PD-L1 positivity was significantly decreased when the PD-L1 threshold was relaxed [15]. Thirdly, the primary site and metastatic lesion may have different expression of PD-L1. This spatial heterogeneity complicates the problem of biopsy site [16]. Fourth, the expression of PD-L1 can be influenced by previous anticancer treatment and suffer from temporal heterogeneity [6,16]. Therefore, recent biopsies should be used to assess the expression of PD-L1. Finally, considering that the PD-L1 status observed on the whole-tissue sample is underestimated by biopsy samples, there may be a relatively poor association of the expression of PD-L1 between the biopsy samples and resected tumors [17]. Therefore, it is very important to understand these limitations of PD-L1 expression when using it as a biomarker for the prediction of ICI efficacy (Table 2).

## 3. Tumor Mutational Burden (TMB) and Response to Immunotherapy

Cancers are caused by somatic mutations that can result in the expression of neoantigens [18]. Notably, TMB affects the generation of immunogenic neoantigens that bring specific T-cell responses. Hence, it is reasonable to hypothesize that TMB influences the response of cancer patients to treatment with ICIs. In fact, it is one of the emerging biomarkers for predicting the response to immunotherapy [19].

Snyder et al. reported that melanoma patients with a clinical benefit from anti-CTLA4 antibody (i.e., ipilimumab or tremelimumab) had significantly higher TMB than those without its benefit [20]. Rosenberg et al. also reported that TMB was significantly increased in responders (i.e., complete response or partial response) versus non-responders (i.e., stable disease or progressive disease) among 150 patients with urothelial cancer who were treated with atezolizumab (12.4 vs. 6.4 mutations per megabase, respectively; *p* < 0.0001) [21]. In a study including 227 patients with advanced NSCLC treated with anti-PD-1/PD-L1 therapies, TMB was significantly greater in patients with a durable clinical benefit (DCB) (i.e., complete response, partial response, or stable disease that lasted >6 months) versus those without a durable benefit. Moreover, patients with TMB > 50th percentile exhibited an improved DCB rate and PFS (i.e., from the date the patient began immunotherapy to the date of progression) versus those with lower TMB (DCB rate: 38.6% vs. 25.1%, respectively; *p* = 0.009) (PFS hazard ratio = 1.38; *p* = 0.024) [22].

In studies that included numerous types of cancers, TMB proved to be a promising biomarker. For example, Goodman et al. reported that higher TMB was associated with a better outcome in 151 immunotherapy-treated patients with 21 cancer types [18]. In this study, the response rate of patients with high (≥20 mutations per megabase) versus low-to-moderate TMB was 58% versus 20%, respectively (*p* = 0.0001); the median PFS was 12.8 months versus 3.3 months, respectively (*p* ≤ 0.0001); and the median OS was not reached versus 16.3 months, respectively (*p* = 0.0036) [18]. Furthermore, in a literature search that included 27 types or subtypes of tumors, a significant correlation between TMB and the objective response rate was observed among patients who had received anti-PD-1/PD-L1 therapies [19].

Currently, numerous clinical trials try to raise the response rates of patients by combining immunotherapies [23]. Of note, TMB may also be a potential biomarker for combination therapy. In 299 patients with advanced NSCLC and a high TMB (i.e., ≥10 mutations per megabase), significantly longer PFS was demonstrated in the treatment with nivolumab plus ipilimumab than that with chemotherapy, irrespective of the expression of PD-L1 (one-year PFS rate: 42.6% vs. 13.2%, respectively) [24]. Hellmann et al. reported that high TMB predicted a better objective response and PFS in 75 patients with NSCLC treated with anti-PD-1 plus anti-CTLA4 antibodies [23]. In this study, TMB was independent of PD-L1 expression. Moreover, it was independently associated with overall response rate (ORR) and PFS in a multivariate analysis. Interestingly, patients with a positive expression of PD-L1 and high TMB showed significantly better rates of ORR and PFS than those with only one or no variable [23]. In another study that included patients with small-cell lung cancer (SCLC) treated with either nivolumab monotherapy (*n* = 133) or nivolumab plus ipilimumab (*n* = 78), better efficacy was observed in patients with high TMB [25].

These results show that TMB is an important biomarker for predicting the efficacy of ICIs. TMB is calculated by the number of mutations per megabase based on next generation sequencing (NGS) technologies using whole exome sequencing or large NGS panels [6]. However, TMB may present temporal variability. Therefore, TMB obtained from a single time point can be insufficient to accurately predict the response to ICIs. For example, in a previous study that included patients with advanced melanoma receiving nivolumab (who had progressed on ipilimumab or were ipilimumab-naïve), pre-therapy TMB was related to OS only in the ipilimumab-naïve group [26]. However, change in TBM after four weeks of therapy (ΔTMB) was strongly correlated with response to nivolumab in the whole cohort [26,27]. Considering that tissue biopsy is an invasive method for the determination of ΔTMB, the use of non-invasive “liquid biopsy” or blood-based sequencing of circulating cell-free DNA (cfDNA) may be favorable [27,28]. Actually, the blood-based assay for the measurement of TMB in the plasma (cfDNA-derived TMB) identified patients with NSCLC who experienced clinically significant improvements in PFS after treatment with atezolizumab [29]. Moreover, we should further investigate the impact of previous chemotherapies or molecular targeted therapies on TMB, in order to assess its usefulness in more accurately predicting the efficacy of ICIs.

## 4. Neoantigens in Cancer Immunotherapy

Tumor-specific mutations can give rise to neoepitopes. These neoepitopes are presented on the surface of tumor cells by major histocompatibility complexes (MHCs). Of note, they may serve as neoantigens (i.e., non-self peptides) that are recognized by T cells [20]. It has been suggested that neoantigens can be a useful biomarker to predict patient response to cancer immunotherapy [30,31]. A study investigated 64 patients with malignant melanoma treated with anti-CTLA4 antibodies. The results showed that specific somatic neoepitopes were shared by patients with prolonged benefits, whereas they were absent in those without prolonged benefits. Moreover, patients with those specific somatic neoepitopes had significantly improved survival versus those without them [20]. Furthermore, in an in vitro validation study, those neoepitopes activated T cells from patients treated with CTLA4 blockade [20]. In other studies, a high candidate neoantigen burden was associated with clinical response and improved PFS among patients with NSCLC treated with an anti-PD1 antibody [32]. In addition, a neoantigen burden was associated with clinical response among patients with metastatic melanoma treated with an anti-CTLA4 antibody [33].

We should also consider the importance of clonal neoantigens (i.e., neoantigens that exist in all tumor cells) and subclonal neoantigens (i.e., neoantigens that exist only in some of the tumor cells), namely neoantigen intratumor heterogeneity (ITH). McGranahan et al. reported that neoantigen ITH influenced the sensitivity of ICIs [34]. In their study, which included 31 patients with advanced NSCLC treated with pembrolizumab, tumors derived from patients without a durable benefit exhibited significantly higher neoantigen ITH than those derived from patients with a DCB. Moreover, tumors with a high clonal neoantigen burden and low neoantigen ITH were correlated with longer PFS. Similarly, among 57 patients with melanoma treated with CTLA4 blockade, significantly improved OS was observed in patients with tumors characterized by a high clonal neoantigen burden and low neoantigen ITH. Furthermore, in those patients with a DCB from ICIs, T cells that recognized clonal neoantigens were detected, indicating that clonal neoantigens possess the ability to promote the neoantigen reactive T cells [34].

Additionally, it has been emphasized that peptide immunogenicity is related with its affinity for MHC. Differential agretopicity index (DAI) is termed as the difference in predicted affinity for any given wild-type and mutant peptide pair, and DAI is an indicator of neopeptide dissimilarity from self [35]. In summary, high DAI values mean that a mutation increases peptide–MHC binding compared with the wild-type sequence, whereas a low DAI means unchanged or weakened affinity for MHC binding [27]. Therefore, we may hypothesize that tumors with high DAI neoantigens are more susceptible to immune recognition by T cells and a clinically relevant benefit induced by ICIs. Actually, low neoantigen DAI was correlated with poorer survival in an immunotherapy-treated melanoma cohort [35]. The relationship between DAI and the efficacy of ICIs should be elucidated in future trials.

Considerable efforts have been exerted to precisely detect candidate neoantigens. Kim et al. developed a machine-learning-based neoantigen prediction program—termed Neopepsee—incorporating nine features of immunogenicity. In independent cohorts of melanoma and chronic lymphocytic leukemia, Neopepsee showed improved performance in the prediction of neoantigens compared with the traditional classification methods. Moreover, this program can be used to identify candidate neoantigens and compare neoantigens with known immune epitopes. This approach may assist in advancing research for the development of next-generation cancer immunotherapies [31].

## 5. The Mismatch Repair Status Predicts the Clinical Benefit of Immunotherapy

Microsatellites are repeat DNA sequences consisting of 2–5 base pairs, occurring 10–60 times. They exist in coding and noncoding regions of the genome [36]. Tumors with genetic defects in mismatch repair pathways have many somatic mutations, particularly in microsatellites. Thus, mismatch repair-deficient colorectal cancers have 10–100 times as many somatic mutations as mismatch repair-proficient colorectal cancers [6]. The accumulation of mutations in these regions of the genome is called microsatellite instability (MSI) [37], and mismatch repair-deficient tumors may activate the immune system. For instance, a previous study has shown that colon cancer patients with increased levels of MSI exhibited higher percentages of cytotoxic T-cell infiltrates versus those with microsatellite stable colon cancers [38]. Another recent study has reported that immune checkpoint ligands (e.g., PD-1, PD-L1, and CTLA-4) are strongly expressed in the tumor microenvironment of mismatch repair-deficient malignancies. This suggests that blockade of these specific immune checkpoints may be effective in cancers with mismatch repair deficiency [39]. Moreover, mismatch repair-deficient tumors have an increased number of mutation-associated neoantigens [37]. Therefore, it is very likely that mismatch repair-deficient tumors are responsive to therapy with ICIs.

In a clinical study, Le et al. reported that colorectal cancers with mismatch repair deficiency were sensitive to anti-PD1 antibody [37]. In this study, 32 patients with progressive metastatic colorectal cancer who had received treatment with pembrolizumab were enrolled (i.e., 11 mismatch repair-deficient cancers and 21 mismatch repair-proficient cancers). The results revealed that patients with mismatch repair-deficient colorectal cancers had significantly better PFS and OS than patients with mismatch repair-proficient colorectal cancers (hazard ratio for disease progression or death: 0.10; *p* < 0.0001, and hazard ratio for death: 0.22; *p* = 0.05). Kim et al. also reported that patients with metastatic gastric cancer and high MSI showed impressive responses to pembrolizumab (ORR: 85.7%) [40]. Another study included 86 patients with advanced mismatch repair-deficient cancers across 12 different tumors. The results revealed that objective radiographic response was observed in 53% of patients, while 21% of patients showed complete response. Moreover, the responses were long-lasting, with a median PFS and OS not reached [41]. On the basis of these results, in 2017, the FDA approved nivolumab and pembrolizumab for the treatment of MSI-positive cancers, irrespective of tumor histology [27]. Yet, larger prospective studies are warranted to confirm the usefulness of the mismatch repair status in predicting the clinical benefit induced by ICIs among various types of cancer patients with high levels of MSI.

## 6. Specific Gene Mutations in Cancer Immunotherapy

The specific gene mutation status may predict the efficacy of ICIs in certain types of tumors. Some of the NSCLC have the driver mutations, including *EGFR*, *STK11,* and *KRAS*, and these mutations are associated with local immune reactions [6]. These mutations of specific genes may affect the ability of tumor cells to escape from immune surveillance [27]. In a retrospective analysis that included 58 patients with NSCLC treated with anti-PD-1/PD-L1 therapies, *EGFR*-mutant or *ALK*-positive patients showed a statistically significant shorter PFS and borderline significant lower ORR compared with *EGFR* wild-type and *ALK*-negative/unknown patients [42]. Rizvi et al. reported that NSCLC patients harboring variants in *EGFR* and *STK11* showed a lack of benefit in the treatment with anti-PD-1/PD-L1 therapies [22]. In addition, a meta-analysis has shown that the *EGFR* mutation status may be a predictive biomarker for OS in patients with advanced NSCLC treated with an ICI (i.e., nivolumab, pembrolizumab, or atezolizumab). The ICIs did not offer an OS advantage in the *EGFR*-mutant patients, whereas there was a 34% reduction in the risk of death in the *EGFR* wild-type patients [43]. On the other hand, *TP53* mutations without co-occurring *STK11* or *EGFR* alterations (*TP53*-mut/*STK11*-*EGFR*-WT) identified the group of lung adenocarcinoma patients with high CD8^+^ T cell density. Moreover, prolonged PFS was observed in lung adenocarcinoma patients with *TP53*-mut/*STK11*-*EGFR*-WT treated with anti-PD1 immunotherapy [44]. Additionally, in their meta-analysis, Lee et al. reported that treatment with ICIs was linked to OS benefit in NSCLC patients with the *KRAS* mutation compared with docetaxel (hazard ratio: 0.65; *p* = 0.03) [45].

Regarding melanoma, an analysis examined 39 patients with metastatic melanoma treated with anti-PD-1 therapies (i.e., pembrolizumab and nivolumab). The findings revealed that patients with PTEN-present tumors showed a significantly better response to anti-PD-1 therapies than patients with PTEN-absent tumors [46]. Another study involving patients with melanoma showed that mutations in either *SERPINB3* or *SERPINB4* were correlated with significantly longer survival following anti-CTLA4 treatment [47].

For RCC, nivolumab showed a clinical benefit among patients with loss-of-function mutations in the *PBRM1* gene, which encoded a subunit of the polybromo and BRG1-associated factors (PBAF) switch-sucrose nonfermentable chromatin remodeling complex. This finding was further demonstrated in a validation cohort of 63 RCC patients treated with anti-PD1/PD-L1 immunotherapies alone or in combination with anti-CTLA4 antibody [48]. Pan et al. reported a reasoning for these clinical results by showing that PBAF-deficient tumor cells are more responsive to T cell-mediated cytotoxicity. They revealed that loss of PBAF function increased tumor cell sensitivity to interferon-γ (IFN-γ), resulting in enhanced secretion of chemokines (i.e., CXCL9 and CXCL10) that recruit effector T cells into tumors [49].

The relationship between specific gene mutations and the efficacy of ICI treatment is summarized in Table 3.

## 7. Gut Microbiome and Anticancer Immunotherapy

Commensal bacteria are useful in maintaining host physiology and immune homeostasis. Therefore, these bacteria are currently attracting attention as a new biomarker of human health [50]. Especially, gut microbiota (weighing approximately 1.5 kg) is the most important type of microbiome in humans [51]. It has been reported that the gastrointestinal mucosa may regulate immune responses at distal organs through the migration of lymphoid cells or inflammatory mediators [50]. Gut microbiota is considered to be an underlying cause of numerous diseases, such as obesity, cardiovascular disease, psychiatric disease, and cancer [51,52]. The gut microbiota affects immunity locally at the mucosal level as well as systemically. Hence, recent studies have focused on its contribution to anticancer immunotherapy [53]. Through the use of probiotics for anticancer immunotherapy or the creation of molecules targeting microbial enzymes, we may expect to improve the efficacy of anticancer therapy [53]. Herein, we will review the relation between cancer immunotherapy and the gut microbiome based on evidence reported by previous studies.

It is established that there is considerable variation among the responses of patients to ICIs. However, previous studies have suggested that the gut microbiota may explain the individual response of patients to treatment with ICIs [53,54]. In a recent study including 249 patients (i.e., 140 patients with NSCLC, 67 patients with RCC, and 42 patients with urothelial cancer) who had received anti-PD-1/PD-L1 therapy, the use of antibiotics within two months prior to or one month after the initial administration of immunotherapy was significantly associated with shorter PFS and OS [55]. Considering that antibiotics can transiently change the composition of the gut microbiome, we can speculate that the gut microbiome may affect the therapeutic efficacy of immunotherapy [55]. In another study, Matson et al. analyzed stool samples from 42 patients with melanoma prior to immunotherapy, and found an association between the microbial composition of the gut and clinical response. In this study, most of the patients were treated with an anti-PD-1 regimen, and it was demonstrated that *Bifidobacterium longum*, *Collinsella aerofaciens*, and *Enterococcus faecium* were more abundant in responders. Moreover, it was revealed that reconstitution of germ-free mice with stool samples from responders led to enhanced T-cell responses, and greater efficacy of anti-PD-L1 therapy [54]. In a preclinical mouse study, it was also confirmed that the oral administration of *Bifidobacterium* alone improved the control of melanoma. Notably, this effect was mediated by the augmented function of dendritic cells, leading to enhanced CD8^+^ T-cell priming and accumulation in the tumor microenvironment [56]. In another study of 43 patients with melanoma undergoing anti-PD-1 immunotherapy, high abundance of the *Ruminococcaceae* family was observed in responding patients [57]. In a larger study including 100 patients (i.e., 60 patients with advanced NSCLC and 40 patients with RCC) who had received anti-PD-1 immunotherapy, metagenomics of patient stool samples at diagnosis revealed correlations between clinical responses to ICIs and the relative abundance of *Akkermansia muciniphila* [55]. Regarding the effects of CTLA-4 blockade, in a mouse model, T-cell responses specific for *Bacteroides thetaiotaomicron or Bacteroides fragilis* were correlated with better efficacy [58]. The gut microbiota associated with good clinical outcome in patients treated with ICIs is summarized in Table 4.

Meanwhile, ICIs are linked to the development of immune-related adverse effects, including immune-mediated colitis. In addition, specific gut microbiota is also reported to correlate with colitis induced by checkpoint blockade [59]. Dubin et al. analyzed the intestinal microbiota of 34 patients with metastatic melanoma undergoing treatment with ipilimumab, and demonstrated that the increased bacteria of the *Bacteroidetes* phylum was correlated with a low frequency of colitis [59]. Thus, fecal microbiota may predict the risk of developing immune-related adverse events caused by ICIs. However, further studies are required to test this hypothesis.

These findings show that the gut microbiome influences the outcome of treatment with ICIs in both mice and humans. However, the microbiota associated with clinical benefit varies between studies. The reasons for these variable results may be different microbial sequencing techniques, and geographic variations in the distribution of gut microbiome [27]. 

An important mechanism of the gut microbiome may be associated with the improved function of tumor-infiltrating effector T cells [54,56,58]. In particular, IFN-γ-expressing CD8^+^ T cells may be vital in antitumor immunity, affecting immune checkpoint inhibitor therapies [60]. Tanoue et al. reported that 11 specific bacterial strains obtained from the feces of healthy human donors act together to induce IFN-γ^+^ CD8^+^ T cells, enhancing the efficacy of ICIs [61]. Further research is necessary to elucidate the therapeutic potential of the gut microbiota inducing IFN-γ^+^ CD8^+^ T cells.

## 8. T Cell-Related Biomarkers

As mentioned earlier in this article, the activity of T cells may play an important role in predicting the efficacy of ICIs. For example, it has been reported that among patients with melanoma, rapid clinical responses to anti-PD1 immunotherapies were correlated with recruitment of exhausted CD8^+^ T cells in the tumor at three weeks [62]. Moreover, Iwahori et al. revealed that the cytotoxic activity of T cells in the tumor was closely associated with that of peripheral blood T cells. They also found that the cytotoxicity of peripheral blood T cells may be a predictor of the efficacy of nivolumab in the tumor microenvironment among NSCLC patients. These results indicate that measurement of peripheral blood T cell activity may predict the immune responses at tumor sites [63].

On the other hand, a previous report demonstrated that CD8^+^ tumor-infiltrating lymphocytes (TILs) possess diverse phenotypes and may be specific for tumor antigens. In addition, they also have a subgroup that identifies cancer-unrelated epitopes, such as those from the Epstein-Barr virus, human cytomegalovirus. This study proposed that CD39 is a possible marker of tumor-specific CD8^+^ T cells, and that CD39^+^ CD8^+^ T cells may predict the efficacy of ICIs [64]. Duhen et al. also reported that CD103^+^ CD39^+^ CD8^+^ TILs showed an antitumor effect in an MHC-class I-dependent manner [65]. They also showed that CD103^+^ CD39^+^ CD8^+^ TILs are related to improved OS in patients with head and neck cancer. Therefore, considering the utility of T cells in predicting the efficacy of ICIs, it is important to identify tumor antigen-specific T cells. 

In another study, it was reported that the response of peripheral blood PD-1^+^ CD8^+^ T cells, measured as the ratio in the percentage of Ki-67^+^ cells seven days after treatment, may predict the response and prognosis of cancer patients treated with anti-PD-1 therapy [66]. Notably, the localization of CD8^+^ T cells may also be important. Among patients with metastatic melanoma, the density of CD8^+^ T cells at the invasive margin of a tumor was more strongly associated with response to pembrolizumab versus central infiltration (i.e., the density of CD8^+^ T cells inside the tumor) [67]. Additionally, a T cell-inflamed gene expression profile was shown to predict response to pembrolizumab among cancer patients [68].

## 9. Conclusions

The results obtained from recent clinical studies have brought cancer immunotherapy into the mainstream of oncology. Although ICIs have been approved for the treatment of numerous types of cancers, thus far, only a small proportion of patients may benefit from their effect. Therefore, numerous studies are currently ongoing to identify biomarkers for the prediction of ICI efficacy. In this review, we focused on the expression of PD-L1, TMB, expression of neoantigens, mismatch repair status, specific gene mutations, and gut microbiome as potential biomarkers of cancer immunotherapy (Figure 1).

Although the expression of PD-L1 appears to be a useful biomarker when using anti-PD-1/PD-L1 therapies, its accuracy in predicting the efficacy of ICIs varies among studies owing to its differing definition, threshold, and spatial/temporal heterogeneity. Moreover, PD-L1 may not be an ideal biomarker in the future because treatment combinations (i.e., chemotherapy plus ICI or a combination of ICIs) are becoming an option for the treatment of cancers. Therefore, we need to identify new biomarkers that can be applied to patients treated with combination therapy. Moreover, the relationship between other biomarkers (TMB, neoantigens, mismatch repair status, specific gene mutations, and gut microbiome) and the efficacy of ICIs should be further elucidated in future clinical trials. We may need to develop a predictive model combining these different biomarkers, which may assist physicians in practicing precision medicine in cancer immunotherapy.

## Figures and Tables

**Figure 1 cancers-11-00935-f001:**
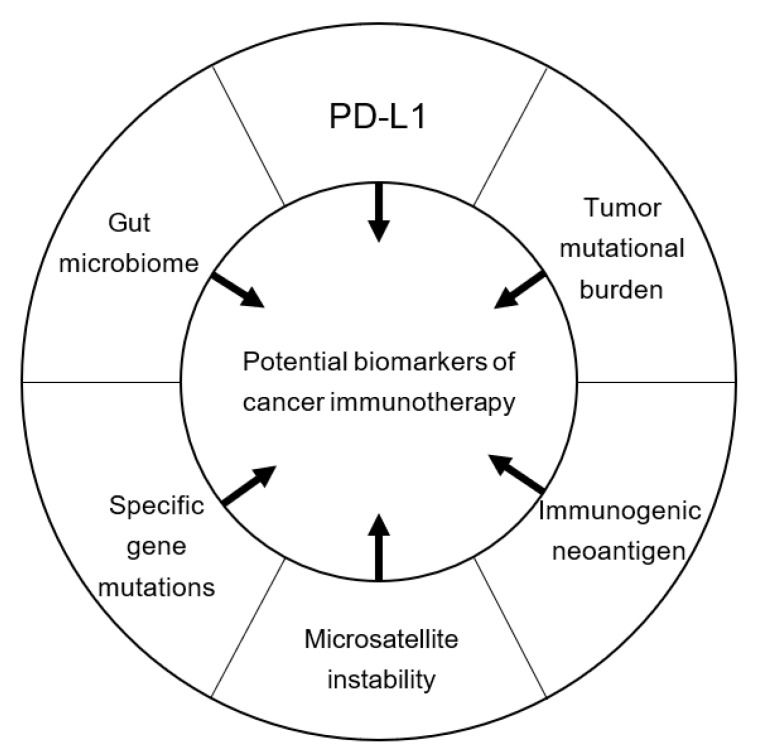
Potential biomarkers of cancer immunotherapy. PD-L1, programmed cell death-ligand 1.

**Table 1 cancers-11-00935-t001:** Association between PD-L1 expression and the effect of anti-PD-1/PD-L1 treatment.

Cancer	Any Association between PD-L1 Expression and the Effect of anti-PD-1/PD-L1 Treatment?	Reference
Melanoma, NSCLC, colorectal cancer, RCC, prostate cancer	Yes	[8]
NSCLC	Yes	[9]
RCC	Yes	[10]
Head and neck cancer	Yes	[11]
RCC	No	[12]
NSCLC	No	[13]

NSCLC, non-small cell lung cancer; PD-L1, programmed cell death-ligand 1; PD-1, programmed cell death 1; RCC, renal cell cancer.

**Table 2 cancers-11-00935-t002:** Clinical reports showing the important limitations of PD-L1 expression as a biomarker.

Limitation	Observation	Reference
Different definitions of PD-L1 expression among reports	Several other studies defined the expression of PD-L1 as the expression on tumor cells. However, this study revealed that the expression of PD-L1 on tumor-infiltrating immune cells, rather than tumor cells, is important.	[14]
No consensus regarding the threshold of PD-L1 expression	This study showed that the predictive efficacy of PD-L1 expression for response to ICIs may depend on its threshold.	[15]
Spatial heterogeneity	In several patients, there were differences in the expression of PD-L1 between primary lung cancers and brain metastases.	[16]
Temporal heterogeneity	In several patients with lung cancer, the expression of PD-L1 may change ≥6 months following the original diagnosis.	[16]
Biopsy specimen	This study showed a relatively poor association of the expression of PD-L1 between lung biopsies and surgically resected specimens.	[17]

ICI, immune checkpoint inhibitor; PD-L1, programmed cell death-ligand 1.

**Table 3 cancers-11-00935-t003:** Specific gene mutations and the efficacy of ICI treatment.

Cancer	Gene Mutations	Efficacy of ICI Treatment?	Reference
NSCLC	*EGFR*	No	[42,43]
	*ALK*	No	[42]
	*STK11*	No	[22]
	*TP53*	Yes	[44]
	*KRAS*	Yes	[45]
Melanoma	*PTEN*	Yes	[46]
	*SERPINB3*, *SERPINB4*	Yes	[47]
RCC	*PBRM1*	Yes	[48]

ICI, immune checkpoint inhibitor; NSCLC, non-small cell lung cancer; RCC, renal cell cancer.

**Table 4 cancers-11-00935-t004:** Gut microbiota associated with favorable clinical outcome in patients treated with ICIs.

Cancer	Gut Microbiota	Reference
Melanoma	Bifidobacterium longum, Collinsella aerofaciens, Enterococcus faecium, and Ruminococcaceae family	[54]
		[57]
NSCLC	Akkermansia muciniphila	[55]
RCC	Akkermansia muciniphila	[55]

ICI, immune checkpoint inhibitor; NSCLC, non-small cell lung cancer; RCC, renal cell cancer.

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
