# Peer review of "Possible Biomarkers for Cancer Immunotherapy"

_cancers, 2019, doi:10.3390/cancers11070935_

Round 1

Reviewer 1 Report

The authors highlighted the potential biomarkers actually available to select patients resposive to immune checkpoint inhibitors. The structure and the rationale of the review are good. However some changes are needed:

1. page 3, lines 91-93: the sentence has to be rewritten because not clear

2. page 4: the authors should describe how TMB is calculated and indicate a reference

3. page 4, lines 145-148: the sentence has to be rewritten because not clear 

4. page 4, lines 165-168: the authors shold better explain this sentence

5. page 5: the auhors should extend the introduction about MSI

6. page 5: a table regarding gene mutations in cancer immunotherapy can help the readers

7. page 6, lines 239-243: the authors should better explain the reasoning mentioned

8. page 7, line 272: the authors should indicate what are the tumors linked to bifidobacterium

9. page 7, lines 276-279 the sentence has to be rewritten because not clear

10. page 7, line 290: the authors should mention what are the biomarkers linked to microbiota

11.page 7, lines 295-299:  the sentence has to be rewritten because not clear

12. the conclusions have to be extended

Author Response

We thank for kind suggestions for our manuscript. Please find our responses below. Based on the valuable suggestions and comments, we have corrected our manuscript. The revised version was created using the track change mode to show the changes made.

Point 1: page 3, lines 91-93: the sentence has to be rewritten because not clear

Response 1: We thank for the comment. In accordance with the suggestion, we have corrected the sentence “Snyder et al. reported that there was a significant difference in TMB between melanoma patients who were treated with an anti-CTLA4 antibody (i.e., ipilimumab or tremelimumab) with a clinical benefit  and those with a minimal or no benefit, both in the discovery set (n=25) and in the validation set (n=39).” to “Snyder et al. reported that melanoma patients with a clinical benefit from anti-CTLA4 antibody (i.e., ipilimumab or tremelimumab) had significantly higher TMB than those without its benefit.”

Point 2: page 4: the authors should describe how TMB is calculated and indicate a reference

Response 2: We thank for the comment. In accordance with the suggestion, we have added the following sentence “TMB is calculated by the number of mutations per megabase based on next generation sequencing (NGS) technologies using whole exome sequencing or large NGS panels [6] ” and cited a reference [6] (lines 131-133 in the revised manuscript).

Point 3: page 4, lines 145-148: the sentence has to be rewritten because not clear

Response 3: We thank for the comment. In accordance with the suggestion, we have corrected the sentences “Moreover, patients with those neoepitope signatures were linked to significantly improved survival versus those without the signatures [20]. Furthermore, in an in-vitro validation study, those neoepitopes activated T cells from patients treated with CTLA4 blockade, and neoantigen-specific T-cell response was demonstrated [20].” to “Moreover, patients with those specific somatic neoepitopes had significantly improved survival versus those without them [20]. Furthermore, in an in-vitro validation study, those neoepitopes activated T cells from patients treated with CTLA4 blockade [20].”

Point 4: page 4, lines 165-168: the authors should better explain this sentence

Response 4: We thank for the comment. In accordance with the suggestion, we have added the following sentence “Differential agretopicity index (DAI) is termed as the difference in predicted affinity for any given wild-type and mutant peptide pair, and DAI is an indicator of neopeptide dissimilarity from self [35].”.

Point 5: page 5: the auhors should extend the introduction about MSI

Response 5: We thank for the comment. In accordance with the suggestion, we have added the following sentence “Thus, mismatch repair-deficient colorectal cancers have 10-100 times as many somatic mutations as mismatch repair-proficient colorectal cancers [6].” in the introduction about MSI.

Point 6: page 5: a table regarding gene mutations in cancer immunotherapy can help the readers

Response 6: We thank for this recommendation. In accordance with the suggestion, we have added a table regarding gene mutations (Table 3 in the revised manuscript).

Point 7: page 6, lines 239-243: the authors should better explain the reasoning mentioned

Response 7: We thank for this recommendation. In accordance with the suggestion, we have corrected the sentence “They also revealed that PBAF-deficient tumor cells produce higher amounts of chemokines (i.e., CXCL9 and CXCL10) by the stimulation of interferon-γ (IFN-γ), resulting in an accumulation of effector T cells into tumors [49].” to “They revealed that loss of PBAF function increased tumor cell sensitivity to interferon-γ (IFN-γ), resulting in enhanced secretion of chemokines (i.e., CXCL9 and CXCL10) that recruit effector T cells into tumors [49].” (lines 249-253 in the revised manuscript).

Point 8: page 7, line 272: the authors should indicate what are the tumors linked to Bifidobacterium

Response 8: We apologize for the lack of information. That is melanoma. We have added it in the sentence.

Point 9: page 7, lines 276-279 the sentence has to be rewritten because not clear

Response 9: We thank for the comment. In accordance with the suggestion, we have corrected the sentence “In a larger study including 100 patients (i.e., 60 patients with advanced NSCLC and 40 patients with RCC) who had received anti-PD-1 immunotherapy, the commensal gut microbiota most significantly associated with favorable clinical outcome in both NSCLC and RCC was Akkermansia muciniphila [55].” to “In a larger study including 100 patients (i.e., 60 patients with advanced NSCLC and 40 patients with RCC) who had received anti-PD-1 immunotherapy, metagenomics of patient stool samples at diagnosis revealed correlations between clinical responses to ICIs and the relative abundance of Akkermansia muciniphila [55].”

Point 10: page 7, line 290: the authors should mention what are the biomarkers linked to microbiota

Response 10: Sorry. The expression “microbiota associated biomarkers” was wrong. We have corrected “microbiota associated biomarkers” to “fecal microbiota” (lines 304-305 in the revised manuscript).

Point 11: page 7, lines 295-299:  the sentence has to be rewritten because not clear

Response 11: We apologize for the confusion. We think that those sentences are unnecessary in this context, and we have removed them.

Point 12: the conclusions have to be extended

Response 12: We thank for this recommendation. In accordance with the suggestion, we have added the following sentence “Moreover, the relationship between other biomarkers (TMB, neoantigens, mismatch repair status, specific gene mutations and gut microbiome) and the efficacy of ICIs should be further elucidated in future clinical trials.” (lines 360-363 in the revised manuscript). Moreover, we have added Figure 1 to summarize the information presented in the conclusions.

Reviewer 2 Report

The article is interesting, comprehensive and well-written. Minor suggestions: 

1-     Line 46-61: A table summarizing the results will be very helpful.

2-     Line 100: Define DCB, PFS

3-     Line 247: gut microbiota weight, please add a reference

4-     The article needs 1-2 figures to simplify and summarize the presented information

Author Response

We thank for kind suggestions for our manuscript. Please find our responses below. Based on the valuable suggestions and comments, we have corrected our manuscript. The revised version was created using the track change mode to show the changes made.

Point 1: Line 46-61: A table summarizing the results will be very helpful.

Response 1: We thank for this recommendation. In accordance with the suggestion, we have added a table (Table 1 in the revised manuscript).

Point 2: Line 100: Define DCB, PFS

Response 2: We thank for the comment. In accordance with the suggestion, we have added the definitions of DCB and PFS.

Point 3: Line 247: gut microbiota weight, please add a reference

Response 3: We thank for the comment. In accordance with the suggestion, we have added a reference [51].

Point 4: The article needs 1-2 figures to simplify and summarize the presented information

Response 4: We thank for this recommendation. In accordance with the suggestion, we have added Figure 1 in the revised manuscript.

Round 2

Reviewer 1 Report

The manuscript can be accepted in present form